# Conditions for Graviton Emission in the Recombination of a Delocalized Mass

**Alessandro Pesci** 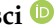

INFN Bologna, Via Irnerio 46, I-40126 Bologna, Italy; pesci@bo.infn.it

**Abstract:** In a known gedanken experiment, a delocalized mass is recombined while the gravitational field sourced by it is probed by another (distant) particle; in it, this is used to explore a possible tension between complementarity and causality in case the gravitational field entangles with the superposed locations, a proposed resolution being graviton emission from quadrupole moments. Here, we focus on the delocalized particle (forgetting about the probe and the gedanken experiment) and explore the conditions (in terms of mass, separation, and recombination time) for graviton emission. Through this, we find that the variations of quadrupole moments in the recombination are generically greatly enhanced if the field is entangled compared to if it is sourced instead by the energy momentum expectation value on the delocalized state (moment variation $\sim m\,d^2$ in the latter case, with $m$ mass, $d$ separation). In addition, we obtain the (upper) limit recombination time for graviton emission growing as $m$ in place of the naive expectation $\sqrt{m}$. In this, the Planck mass acts as threshold mass (huge, for delocalized objects): no graviton emission is possible below it, however fast the recombination occurs. If this is compared with the decay times foreseen in the collapse models of Diósi and Penrose (in their basic form), one finds that no (quadrupole) graviton emission from recombination is possible in them. Indeed, right when $m$ becomes large enough to allow for emission, it also becomes too large for the superposition to survive collapse long enough to recombine.

**Keywords:** quantum nature of gravity; graviton emission; collapse models

## 1. Introduction and Background

To date, there is still no direct evidence for a nonclassical nature of the gravitational field. Quantum effects accompanying gravity are expected to unavoidably show up at the Planck length scale $l_p$. Many of the proposed tests on quantumness of gravity involve consideration of cosmological or astrophysical circumstances, in which the cumulative effects over long distances might compensate for the smallness of $l_p$. A trouble with this is the lack of full control of the experimental circumstances, i.e., our degree of ignorance/uncertainty concerning the model of the universe, the source, and the propagation of the signal to the observer.

The alternative is laboratory tests on systems suitably designed to let potential quantum features of gravity to show up, following a proposal proposed originally by Feynman. The idea is [1] that the final quantum state of a system in which a delocalized mass is allowed to gravitationally interact with another mass ought to be different depending on whether the mediating field is quantum or classical.

The difference appears very hard to detect, but advances in quantum technologies have, by now, made these kind of tests feasible, or at least conceivable in practice. Quantum systems are used as sources of the field typically in a superposition of locations ([2,3] for review and discussion), the main difficulty to face in this kind of effort being decoherence [4].

Suggestions have been made, for example, to look at stochastic fluctuations of quantum origin in the gravitational field [5–7]. Starting from [8,9], a new twist has been given to the subject with the proposal to directly check quantum coherence aspects of gravity, in the form of the ability of the gravitational field to entangle systems initially prepared in

a separable state. The point is that no entanglement can be created by two parties that communicate exclusively through a certain local channel if the latter is classical [10,11]; the appearance of entanglement between two accessible parties from initial conditions of no entanglement would then exhibit a nonclassical nature for the mediating unaccessed channel [12].

Strictly speaking, we can argue that the communication processes involving the gravitational field might be nonlocal, yet causal [13]. If this is the case, there would be no mediators and the just-mentioned creation of entanglement would prove that quantum sources do create superpositions of geometries, yet without gravitons to mediate this [13]. In this paper, we assume the locality of the gravitational channel, since our focus is on the possible emission of (physical) gravitons in the recombination of a delocalized source.

The proposals [8,9] have suggested to consider two masses, each one delocalized, interacting exclusively through their gravitational field. The masses are prepared in a separable state, allowed to interact gravitationally, and are eventually tested for entanglement. The experimental requirements accompanying these kind of tests place their feasibility in a hopefully not-so-far future. This possibility appears even closer when looking at [14]. In it, an experimental setup is considered in which the strength of the gravitational interaction is increased through use of a very heavy (not delocalized) mass, which acts as a mediator between an unlocalized mass and an ancillary qubit.

Building on Feynman's observation, other circumstances can be considered, in which, even leaving the actual feasibility apart, the difference between the effects of quantum versus classical mediating gravitational field can be evident and possibly rich in consequences at the theoretical level. One example, which is our starting point here, is the configuration described schematically in Figure 1 [15,16]. In it, a particle, that we call Alice's particle $A$ with mass $m_A$, is held (from a distant past) in a superposition of locations (paths 0 and 1 with separation $d$), and another particle, $B$ at a distance $D$, Bob's particle with mass $m_B$, (only) gravitationally interacts with $A$. At a preassigned time, Alice starts recombining $A$ and Bob releases $B$ (or decides not to). Alice will perform her task in a time $T_A$, and Bob will check for the position of $B$ after a time $T_B$ from the release (we assume the experiment is local, with Alice and Bob having no relative motion and sharing a local frame). If the gravitational field indeed entangles, the superposed positions of $A$ are accompanied by different fields at $B$ (the two locations of $A$ give rise to different quadrupole moments for Alice's system, and the gravitational field at $B$, entangled with $A$'s locations, is in a superposition of the two states sourced by the corresponding quadrupole moments), and Bob can in principle be able, after a certain minimum time $T_{\mathrm{wp}}$, to discriminate between the paths of $A$.

The configuration we are describing was proposed in [15] (which elaborated on a previous investigation on gravitational tagging of the path [17]) then reconsidered in [16], and further discussed in [18–21]. In these works, this is viewed similar to the scene of a gedanken experiment, the focus being on describing a seemingly paradoxical situation arising from requiring both the complementarity principle—meant as the fact that obtaining which-path as performed by Bob must be incompatible with Alice being able to recombine coherently—and causality. In particular, the perspective in [15] is to extract from the avoidance of a potential paradox the existence of a minimum time Alice needs in order to find if the state of $A$ is a coherent superposition or a mixture.

The premise for the arising of a paradox is, as mentioned, the assumption that the gravitational field at Bob's location can possibly allow for discrimination of the path of $A$. If this is the case, and if circumstances are such that the distance $D$ between $A$ and $B$ is larger than $T_A, T_B$ (we use Planck units through all the paper unless explicitly stated otherwise), then the which-path Bob performs apparently leads by complementarity principle to superluminal transmission of information from Bob to Alice ($A$ has to lose coherence). If, on the other hand, the gravitational field at $B$ cannot distinguish the path, as would be the case if the field is sourced by a mixture of the two paths, then no paradox at all can arise (cf. [21]). The latter is, for example, the case if the gravitational field at $B$ is sourced by

the expectation value $\langle T_{ab} \rangle$ of the energy–momentum tensor of *A* (and its lab) (this would be gravity in its semiclassical description, matter is quantum but the gravitational field is classical): in this case, the gravitational field feels a mixed state of paths, and the positions of *B* are not entangled with the single possible paths. This makes it clear that the assumption of the gravitational field being able to entangle, that is (with locality assumption), of being quantum, is at the origin of the possible paradox.

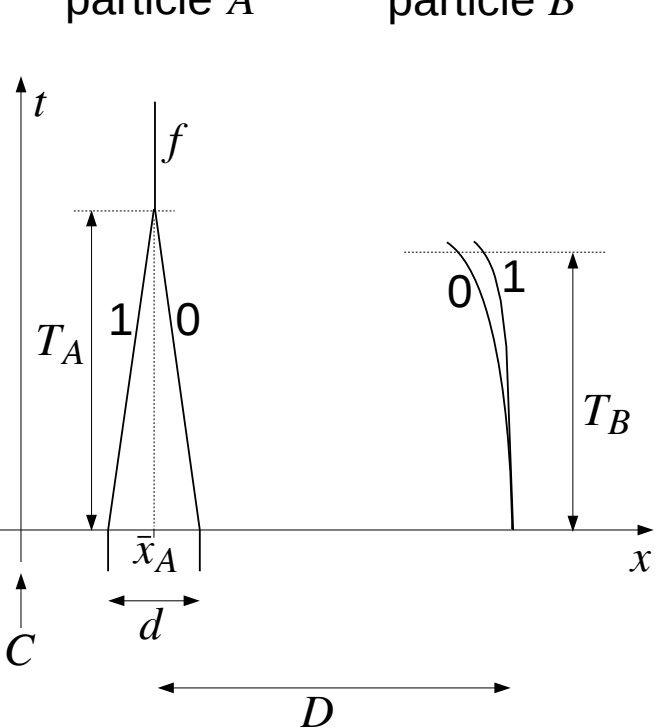

**Figure 1.** Setup of the thought experiment [15,16] as used in the analysis here (see main text). The *x* coordinate is distance taken from Alice's system's center of mass *C* (lab + her particle *A*), and *C*'s worldline acts as the time axis. At a same time ($t = 0$), Alice starts recombining *A*, from a (held from long before) superposition of locations (with separation *d*), and Bob releases his particle *B* located at a distance $D \gg d$ from *A*. Alice completes her task in a time $T_A$ while Bob checks the position of *B* at $t = T_B$. The labels 0 and 1 tag the superposed configurations of the system (no superposition for particle *B* in case gravity is not able to entangle). *f* tags *A* when it is undelocalized, assuming it is located at a small distance $\bar{x}_A$ from *C*.

According to [16,18,19], the overcoming of the paradox is in the interplay between the spatial resolution, unavoidably finite, of Bob in determining the position of *B* (ideally the Planck length $l_p$), and the fact that when Alice recombines *A* quickly enough, Alice's system emits gravitational radiation (from the variation in the quadrupole moment of Alice's system) in the form of a quantum of radiation, namely, a graviton. In practice, were circumstances (read, the difference of quadrupole moment of Alice's system for the two positions of *A*) such that Bob would be able to obtain which-path with $T_{wp} < D$, then, in case $T_A < D$, *A* would necessarily be above the threshold for graviton emission. That is, the coherence of *A* becomes destroyed regardless of what Bob actually does, since we see that happens even if Bob decides not to release the particle. In this, they then move one step farther with respect to [15], in that they do not only recognize the existence of a limit time in performing coherently the recombination, but also they identify the underlying reason for it. The absence of a paradox comes, then, as a consequence.

Here is the point of the present investigation. The emission of quadrupole radiation is clearly conceivable only if the quadrupole moment of Alice's system changes in the recombination. This is precisely what one would expect if the gravitational field has values

entangled with the superposed positions of *A*. One might then think of the emission of radiation by Alice's system during recombination of *A* as a way to tag the ability of the gravitational field to become entangled with the path, regardless of any possible recourse to a test particle *B*. In the case of the classical gravitational field, sourced by the expectation value of the energy–momentum tensor on the delocalized state, we also expect a variation of quadrupole moment in the recombination. This variation, however, turns out to be generically negligibly smaller than the above, as we will see. This aside, the emission of (classical) radiation would originate from the delocalized state as a whole and we would not expect it to affect the coherence of *A*.

In principle, we could then think of an experiment in which the quantum nature of the gravitational field might be checked, under locality assumption, using only one delocalized mass: Alice's particle *A* here, looking at that when it is recombined quickly enough—in a time below a certain threshold $T_A < T_{\text{emit}}$—it emits a graviton, this being witnessed by the abrupt loss of coherence in an ideal situation in which the environmental influence on *A* is taken under full control. One can guess that an experiment of this kind is similar to an impossible task. However, leaving any actual feasibility aside, there might be, from a theoretical point of view, an interest in having a closer look at the conditions one has to require to allow for graviton emission, and this is the aim of the present work. As a byproduct, some indications on the (im)practicability of such an experimental scheme will also emerge, as we will see (as well as some specifications about the actual reasons behind the avoidance of the paradox).

## 2. Conditions for Graviton Emission

Let us start from the analysis in [16,18,19], which is within the just-discussed approximations and limits. With reference to Figure 1, it is shown that in case the gravitational field felt by *B* is entangled with the path of *A*, then, assuming the spatial resolution is limited by the Planck length $l_p$, Bob cannot perform which-path in a time $T_B < T_{\text{wp}}$ with

$$T_{\text{wp}} \sim \frac{D}{\sqrt{Q_A}} \, D; \tag{1}$$

On the other hand, during recombination of *A*, Alice's system will emit (at least) a graviton if $T_A < T_{\text{emit}}$ with

$$T_{\text{emit}} \sim \sqrt{Q_A}. \tag{2}$$

In these equations, $Q_A$ is assumed to be the order of magnitude of both the difference (taken positive) between the quadrupole moments of Alice's system for configurations 0 and 1 and for before and after recombination of *A* (at leading order we have a quadrupole term, not dipole, since the dipole contribution is suppressed by momentum conservation of Alice's system [16]). From these results, we see that whenever Bob can actually perform which-path in $T_B < D$, this from (1) means that we must have $D < \sqrt{Q_A}$, and then if $T_A < D$, necessarily $T_A < \sqrt{Q_A}$, and Alice's system emits [16,18,19]. No paradox can then arise.

The actual value of $Q_A$ is of no importance in the argument above. The authors of [19] take this to be

$$Q_A = d^2 \, m_A, \tag{3}$$

as might be envisaged by dimensional considerations. In what follows, we will see that generically $Q_A$ might be actually expected to be quite larger than this, and that the discrepancy might be of importance as a signature of the emission being nonclassical.

We proceed now with our analysis. We are working under the assumption that the gravitational field requires a quantum description, and the field sourced by *A* is entangled with the superposed positions. We assume then that the energy–momentum densities

$T_0^{ab}$, $T_1^{ab}$, $T_f^{ab}$ ($a, b$ are spacetime indices) associated with the configurations 0, 1, and $f$ of Alice's system do quantum correlate with the gravitational field they generate, described quantum-mechanically by the field states $|\phi_0\rangle$, $|\phi_1\rangle$, and $|\phi_f\rangle$ respectively, and we write the state of Alice's system ($A$ + fields) as

$$|\psi\rangle = \frac{1}{\sqrt{2}}\left(|x_0\rangle|\phi_0\rangle + |x_1\rangle|\phi_1\rangle\right), \tag{4}$$

at $t = 0$, and

$$|\psi_f\rangle = |x_f\rangle|\phi_f\rangle, \tag{5}$$

at $t = T_A$, where $|x_0\rangle$, $|x_1\rangle$, and $|x_f\rangle$ are the states of $A$ corresponding to configurations 0, 1, and $f$, namely, with the center of mass of $A$ at coordinates $x_0$, $x_1$, and $x_f$ where these are taken in the center of mass frame of Alice's system which we choose as our reference frame. As far as the two positions of $A$ are well separated ($\approx$ no overlap between the states of $A$ describing each given position), the matter states can be taken as orthogonal. As for the field states, in the linearized quantum theory of gravity ($g_{ab} = \eta_{ab} + h_{ab}$ with $\eta_{ab}$ the Minkowski metric and perturbation $h_{ab}$ small) which we find appropriate in our circumstances, we assume we can proceed analogously to the electromagnetic case (with energy–momentum tensors replacing currents, cf. [22]). In the latter case, the overlap $\langle\varphi'|\varphi\rangle$ between two states $\varphi'$, $\varphi$ associated with the currents $j'^a$, $j^a$ can be written as the overlap between the vacuum and the state associated with the current $j^a - j'^a$ [23,24]. Here, we have $\langle\phi_1|\phi_0\rangle = \langle 0|\phi_0 - \phi_1\rangle$, with $|\phi_0 - \phi_1\rangle$ the field state generated by $T_0^{ab} - T_1^{ab}$. The differences $T_0^{ab} - T_f^{ab}$, $T_1^{ab} - T_f^{ab}$ are what is expected to produce radiation (the entangling part coming from the difference of the two, $T_0^{ab} - T_1^{ab}$, cf. [16]). These differences might be generically expected to be quite significant in the region of superposition, thus giving $|\phi_0\rangle$ and $|\phi_1\rangle$ nearly orthogonal. It is assumed, however, that the two components, even if nearly orthogonal at $t = 0$, can be fully recombined in time $T_A$ [16]. We will return to this later.

As for the emission during recombination, we are free to probe the possibly emerging radiation where we like. We use this freedom, imagining to analyze it at distances much larger than $d, \bar{x}_A$; $d \ll D$ is also the condition used in [16,18,19]. Assuming there is an analogy with classical emission, this enables us to describe the radiation in terms of multipole expansion in powers of $1/r$, having taken $r$ the distance from the center of mass $C$ of Alice's system, and in practice approximate it with the lowest-order terms. What matters are the differences between the configurations 0 or 1 and $f$. Assuming momentum (and angular momentum) conservation during recombination, the dipole term gives vanishing contribution, and the emission is determined by (the third time derivatives of) quadrupole moments $Q$. We have to consider the differences $Q(0) - Q(f)$, $Q(1) - Q(f)$, and $Q(0) - Q(1)$ (with obvious meaning of the notation), with the latter being the part responsible for emission of entangling radiation (in this case, the field emitted is entangled with the source and then brings decoherence and fading out of the final interference pattern) [16]. The latter quantity is also clearly responsible for the difference of the two superposed fields felt by a probe at $r$ (as, in particular, $B$).

In view of our task of determining the conditions for graviton emission, what we perform here is simply computing these differences of quadrupole moments, having in mind a situation generic with $A$ delocalized starting from a position not necessarily coinciding with $C$, with coordinate $\bar{x}_A$ (which we take as non-negative) with respect to $C$ taken as origin.

We find (the calculation is spelled out in the Appendix A) that, denoting $2Q_A \equiv Q(0) - Q(1)$, $Q_A$ turns out indeed to be also $Q_A = Q(0) - Q(f) = |Q(1) - Q(f)|$, at least when $\bar{x}_A$ is significantly larger than $d$, and

$$Q_A = 2m_A\bar{x}_A d = \frac{2\bar{x}_A}{d}m_A d^2. \tag{6}$$

In particular, a same $Q_A$ thus rules emission and entangling emission (and which-path discrimination at distance $r$).

As for the avoidance of the paradox along the lines of [16,18,19], this changes nothing for it amounts merely to replace (3) with (6) in giving $Q_A$ (assuming $D \gg d, \bar{x}_A$). It just confirms that we can use a same $Q_A$ in (1) and (2).

We see that (6) gives a dependence on the quantity $m_A d^2$, which is $Q_A$ in (3), with a factor $2\bar{x}_A/d$ in front. Both give, then, a threshold time for emission which goes like $\sqrt{m_A}$. The factor $2\bar{x}_A/d$ in (6) can give, however, a value for $Q_A$ in principle much bigger than (3), when $\bar{x}_A$ is significantly larger than $d$, and this has consequences for the limit time. From (2) and (6), we obtain

$$T_{\text{emit}} \sim \sqrt{Q_A} = \sqrt{\frac{2\bar{x}_A}{d}}\sqrt{m_A}\,d = \sqrt{\frac{2\bar{x}_A}{d}}\sqrt{\frac{m_A}{m_p}}\frac{d}{c}, \tag{7}$$

where, in the last equality, we reinserted all constants with $m_p$ the Planck mass and $c$ the speed of light in vacuum.

In the first step here (which is Equation (2)), we are making a classical analogy and taking $\frac{1}{T_A}$ as basic, characteristic scale of the angular frequency $\omega$ of the emerging radiation [16] (similar to if recombination were obtained harmonically or similar to in the emission part of Thomson scattering). The emitted power is $\sim(\dddot{Q}_A)^2$ (dots are time derivatives) and the emitted energy $\mathcal{E}$ in time $T_A$ can be written as

$$\mathcal{E} \sim \int_0^{T_A} \dddot{Q}_A^{\,2}\, dt = \dddot{Q}_A^{\,2}\, T_A, \tag{8}$$

with $\dddot{Q}_A \sim \frac{1}{T_A}\left(\frac{1}{T_A}\frac{Q_A}{T_A}\right)$. If $\mathcal{E}$ is in gravitons of energy $\frac{1}{T_A}$, the emission of at least one of them is possible indeed only if $T_A < \sqrt{Q_A}$ [16]. Emission at lower frequencies can also be expected both in a classical (Fourier transforming a generic pulse of duration $T_A$, angular frequencies up to $\frac{1}{T_A}$ are significantly present) and a quantum setting (population of low-energy states in the radiation field), even if depressed (radiated power is $\omega^6$ for a quadrupole source) and of lower impact in reducing the coherence of $A$. This means that we have a tail of emission also when $T_A > \sqrt{Q_A}$, but the gross picture is that there is an effective threshold recombination time of a characteristic scale $\sim\sqrt{Q_A}$ to have emission; this is what we refer to when considering the conditions for the onset of emission.

This analysis is for when the gravitational field does entangle with the superposed locations. This corresponds to having the field state at a point as a superposition of (nearly orthogonal) states $|\phi_0\rangle$ and $|\phi_1\rangle$, as said, and gives $\approx\pm Q_A$ as variations of quadrupole moments associated with each of the branches of the superposition and $\approx 2Q_A$ as difference of quadrupole moments between the branches. In case the field does not entangle with the superposed locations and is sourced instead by the expectation value of the energy–momentum tensor on the delocalized state, we have a single expression for the field $\phi$ given as the gravitational field sourced by the mass density distribution:

$$\rho(x^i) = \frac{1}{2}\left[m_A\delta(x^i - x_0^i) + m_A\delta(x^i - x_1^i)\right] + \rho_l \tag{9}$$

(describing $A$ in terms of Dirac's $\delta$) where $x_0^i = (x_0, 0, 0)$ and $x_1^i = (x_1, 0, 0)$ are the two superposed positions of $A$, and $\rho_l$ is the mass density distributions of Alice's lab, $A$ excluded. In this case, we cannot have a difference connected with positions 0 and

1 (energy–momentum densities of both positions contribute to the field, no which-path possible), and the variation $\widetilde{Q}_A$, and also the emitted energy, we obtain in the recombination is much smaller than in the case that the field is entangled. Indeed, as can be easily verified (cf. Appendix A), $\widetilde{Q}_A = \frac{1}{2} d^2 m_A = \frac{1}{2} Q_A / (\frac{2\bar{x}_A}{d})$ which is $\ll Q_A$ for $d \ll \bar{x}_A$. We see that the factor $\frac{2\bar{x}_A}{d}$ gives the order of magnitude of $Q_A / \widetilde{Q}_A$, the gain we have in the amplitude if the field entangles, and $(\frac{2\bar{x}_A}{d})^2$ the gain in emitted energy.

The factor $d/c$ in (7) clearly has the meaning of absolute lower limit to the recombination time $T_A$ for two paths separated by a distance $d$: $T_A \geq d$ always (cf. [15]). In addition, if the recombination takes places in a time $T_A$, we have to consider that only a portion of size $cT_A$ (inserting, explicitly, the speed of light) of Alice's system can be involved in momentum transfer (this being the part involved in overall momentum conservation during the recombination of $A$), as components of Alice's system far from one another and from $A$ by more than this distance have no time to talk each other in reaction to $A$ recombination.

This suggests that when following the approach of [16,18,19], one has to be careful with the definition of the system under examination. In particular, we have to consider the consequences of that; along with [16,18,19], we *assume* that $A$ can be coherently recombined in time $T_A$, namely, with the fields following the matter evolution from the superposed state (4), with nearly orthogonal $|\phi_0\rangle$ and $|\phi_1\rangle$, to the recombined state (5). Indeed, the gravitational field at points farther than $cT_A$ from particle $A$ cannot be causally affected by Alice system's evolution during recombination, and then the assumption that the superposition can be coherently recombined requires, for consistency, that only that same portion of size $\approx cT_A$ of the system is involved. This would correspond to the view that the local physics of $A$ and its neighborhood ought to be ruled in terms of quantities causally connected with it. Alice's system can thus be considered as effectively made by this portion with particle $A$ and the fields there, and $C$ ought then to be taken as the center of mass of this reduced system. This implies that we always have $T_A > 2\bar{x}_A$ in Equation (7).

The latter condition is the only remnant of this local physics request. Looking at the Appendix A, we see, indeed, that all the calculation carries over for this restricted system provided the reduced system mass $M'$ is still $\gg m_A$, and gives the same results (A8) and (A9), with $Q_A$ given again by (6).

Concerning the gedanken experiment, for $T_A < D$ (which is the interval we are interested in), this, strictly speaking, leaves out $B$ and the region around it. This is not a real problem, however, since, from the considerations just made, the (differences of the) fields felt by $B$ from the restricted configuration (fields suitably extended till reaching $B$) are practically the same as those actually sourced by the full mass distribution of Alice's system.

A tension between the request that Alice is able to recombine $A$ in a finite time $T_A$ and the fact that the fields do extend far beyond $cT_A$ and cannot keep up with the evolving source was emphasized in [25]. There, it was argued that this can be taken as showing that, if $A$ can actually be coherently recombined in time $T_A$ and thus is able to provide interference patterns, then the field states at $t = 0$ ought not to be nearly orthogonal, but instead have sizable overlap. Building on this, Ref. [25] showed this would imply that the interference fringes, if present in the absence of a test body (this meaning the fields produce negligible decoherence), would never disappear, whichever body might interact with the fields (in particular, our test particle $B$). This eliminates any possibility of superluminal communication with $A$ and would solve, once and for all, any potentially paradoxical issue in the gedanken experiment. One might feel, however, a little uncomfortable with the fact that in this approach, the gravitational fields sourced by a delocalized (and then recombined) particle, such as, for example, those in the full-loop Stern–Gerlach interferometers envisaged in the proposals [8,9], apparently would be far from being orthogonal to each other even if sourced by orthogonal states.

Here, we take quite a complementary view. On the basis of the fact that the physics of $A$ should also be describable in terms of only local quantities, namely, quantities able to affect $A$ in time $T_A$, we maintain that if Alice's system (including its fields) is effectively restricted to a causally connected neighborhood of $A$, we can actually have coherent recombination

starting from orthogonal $|\phi_0\rangle$, $|\phi_1\rangle$, thus having orthogonal field states corresponding to orthogonal source states. We stick, thus, to the protocol [16,18,19], only noting that it requires a redefinition of the system effectively taking part in the action. This might not seem to solve all potential issues (cf. [25]) accompanying the gedanken experiment in the protocol [16,18,19], but shows at least the viability of coherent recombination of orthogonal states in the mentioned restrictions. This is all we need here after all, our focus being not an examination of the gedanken experiment, but, instead, the possible graviton emission, and the latter is dictated by the variations $Q(0) - Q(f)$, $Q(1) - Q(f)$, which do not depend, as discussed, on the approach.

This said, we search now for conditions which allow for graviton emission. For this, we must have

$$d < T_A < \sqrt{Q_A}, \tag{10}$$

which is

$$1 < \frac{T_A}{d} < \sqrt{\frac{m_A}{m_p}} \sqrt{\frac{2\bar{x}_A}{d}}, \tag{11}$$

inserting explicitly the Planck mass.

Inequalities (11) give

$$1 < \frac{T_A}{d} < \sqrt{\frac{m_A}{m_p}} \sqrt{\frac{T_A}{d}}, \tag{12}$$

which is clearly impossible to satisfy as long as $m_A < m_p$, for $\frac{T_A}{d} > 1$ implies $\frac{T_A}{d} > \sqrt{\frac{T_A}{d}}$. That is, if $m_A < m_p$, we can never have $T_A < T_{\text{emit}}$, i.e., graviton emission associated with recombination, and this is regardless of the choice of $\bar{x}_A$. The Planck mass acts as a lower-limit threshold mass $m_{\text{emit}}$ for quadrupole emission, the latter being possible only if $m_A > m_{\text{emit}} = m_p$.

Present technology, and that foreseen in the near future, gives a delocalized $m_A \ll m_p$ by far. Alice's (thought) experiment on $A$ (as well as the action on each of the two delocalized particles in actual experimental proposals [8,9] checking for the nonclassical nature of gravity) is akin to completing a Stern–Gerlach apparatus with a recombination stage to obtain a proper Stern–Gerlach interferometer. A first realization of such a device was recently reported at single-atom level [26], with possible extensions of this same experimental procedure to nanodiamonds ($10^6$ carbon atoms, $m_A \approx 10^{-20}$ kg) appearing within reach. In addition, for microdiamonds of $m_A \approx 10^{-14}$ kg (radius $\approx 1$ μm), coherence times of $\gtrsim 1$ s might be conceivable under cooling [27], and delocalizations of objects of this mass with separation of order of their size might be within reach soon [28]. These figures are expected to be good enough for proposals [8,9] to start to be effective, but, anyway, leave $m_A \ll m_p = 2.18 \cdot 10^{-8}$ kg.

Looking at present and near-future capabilities, we thus cannot have graviton emission associated with recombination even if $T_A$ is taken as short as causally allowed. However, from a theoretical point of view, we can be allowed to imagine full-loop Stern–Gerlach interferometers working with $m_A > m_p$. In them, $A$ can be recombined fast enough to allow Alice's system to emit. In (11) (right inequality), we see that the threshold time $T_{\text{emit}}$ depends on $\bar{x}_A$. The best option for allowing emission for a given $T_A$ is to have $\bar{x}_A$ as large as possible, namely, such that $2\bar{x}_A = T_A$. We assume that this choice is not only possible if $T_A$ is just above $d$ but that it is generically realized with Alice's system being macroscopic. With it, inequality (11) coincides with (12) and allows that we have emission when

$$T_A < \frac{m_A}{m_p} d, \tag{13}$$

that is,

$$T_{\text{emit}} \sim \frac{m_A}{m_p} d, \tag{14}$$

which can also be derived from (7), taking $2\bar{x}_A = T_{\text{emit}}$. We see that the condition of emission depends this way generically on parameters concerning particle $A$ alone ($m_A, d$) as one might have hoped, not on Alice's lab. Notice that inequality (13) gives a threshold time which grows linearly with $m_A$, not as $\sqrt{m_A}$, as might seem to be inferred instead from (7).

This as far as the ability of Alice's system to emit is concerned. Regarding, instead, the paradox, notice that its avoidance when $m_A < m_p$ is in that for these masses we cannot have (by far) $\sqrt{Q_A} > D$, and thus Bob cannot perform which-path in $T_B < D$ in the first place. Indeed, from (6) (with $m_p$ inserted), we have

$$\sqrt{Q_A}/D = \sqrt{2\bar{x}_a/D}\sqrt{m_A/m_p}\sqrt{d/D}, \tag{15}$$

which clearly is $\ll 1$ for $m_A < m_p$ if $\bar{x}_A, d \ll D$.

Generic $m_A > m_p$ is still not enough for the potential onset of the paradox. In view of (15), we have to indeed require $m_A \gg m_p$ in order to have $\sqrt{Q_A} > D$. When $m_A$ is large enough to give this, Alice's system necessarily emits [16,18,19], as described above, and no paradox can in anyway arise.

Inequality (13) coincides with the mentioned minimum discrimination time reported in [15] (Equation (3) in [15]) needed to avoid the paradox, in spite of being (quite unconvincingly, cf. [16]) derived there from consideration of dipole gravitational moments (that is, neglecting the reaction of Alice's lab to the displacements of particle $A$, a reaction which brings instead to momentum, and thus dipole moment, conservation); notice, however, that according to our results, the no-paradox argument used in [15] can be leveraged only when $m_A \gg m_p$, as just mentioned.

Further, if we imagine that Alice checks the coherence of particle $A$ through an interference experiment (as considered in [16,17,20]), the minimum allowed time to have the fringes ideally discernible (on account of the finite spatial resolution limit $l_p$) does coincide with the threshold time (14) for emission. Indeed, following [20], if we call $\delta$ the fringe spacing, we have (with all constants) $\delta \sim \lambda v T_A/d \sim l_p \frac{m_p}{m_A} \frac{cT_A}{d}$, where $v$ is the velocity of $A$, $\lambda = h/(m_A v)$ its de Broglie wavelength, and $h$ is (unreduced) Planck constant. From this, requiring that $\delta > l_p$, we obtain (13).

This finding of equivalence/coincidence between no-emission condition and (ideal) detectability of fringe pattern in an interference experiment is at variance with [20], where (using (2) with $Q_A$ given by (3)) the visibility of fringes is found to constrain more than no-emission (when emission sets in, the fringes are undetectable already). However, when $\bar{x}_A$ is not maximal (i.e., when $2\bar{x}_A < T_A$, quite a nongeneric situation as we mentioned) we also find, as per [20], that when emission sets in, fringes' visibility is already lost. The general picture we obtain is that emission has all that is needed to avoid the paradox, but fringes' discernibility taken alone (i.e., without considering emission) is also fine for this; moreover, in generic circumstances, the two requirements do coincide. They are then basically equivalent concerning the avoidance of the paradox in an interferometric setup.

This confirms the stance [20] that, at least as far as checking of coherence of $A$ is carried out through interference, the limit posed by existence of a limit length $l_p$ is enough (without, strictly speaking, a need of bringing into play emission, but being, as we find here, equivalent to the no-emission condition) to avoid any clash between complementarity and causality. This is also what [17] found (though neglecting there, too, the abovementioned reaction of Alice's lab to the displacements of $A$, i.e., using dipole gravitational momenta).

In [19], however, a different setting to probe the coherence of $A$ is considered, not relying on the detection of an interference pattern. Looking at this, it seems we have inevitably to require graviton emission to avoid the paradox in case $m_A \gg m_p$ (clearly,

provided gravity is supposed to be able to entangle; if not, no paradox can arise). This occurs when supposing that locality holds. Assuming, instead, nonlocality of the gravitational communication channel (as contemplated in [13]), it is not clear how to avoid the paradox (when $m_A \gg m_p$) since we have, of course, causality anyway and no quantized mediators to react on $A$ which is causally disconnected from $B$; but we will return to this in the final comments of the paper. The consideration of the potential paradox might highlight a possible weakness of (causal) nonlocality of the channel as compared to locality.

### 3. Contrasting with Collapse Models

In (7) and (14), the Planck mass $m_p$ plays a pivotal role in that it sets the mass threshold for particle $A$ to emit. In particular, these expressions say, as discussed, that if we have a delocalized particle we cannot expect quadrupole emission on recombining it if $m_A < m_p$.

We would like to ask now how this compares with Diósi's and Penrose's hypothesis [29–31] that any such superposition of a mass $m$ in two locations is unstable when the mass is large, and collapses or decays to one of the two locations with average lifetime $\tau = \hbar/E_\Delta$ (all constants in), where $E_\Delta$ is the gravitational self-energy of the difference of mass configurations in the two locations, up to a multiplicative constant ([32] for details, see also [33]). As a matter of fact, this model seems ruled out in its basic formulation [34], but there is still a dependence on some parameters. We ask for which masses $m$ the decay time $\tau$ keeps being large enough to allow for quadrupole emission from recombination if $T_A$ is taken sufficiently short.

For this, we take the expression $\tau = \frac{5}{6} R/m^2$ of [32] for a uniform massive delocalized sphere with radius $R$, valid when the separation is $d \gg R$ and for a specific/reasonable choice of the multiplicative constant (given by the parameter $\gamma$ in [32] set to $\frac{1}{8\pi}$, which is $\sim$50 times smaller than the value ruled out in [34], thus giving $\sim$50 times longer lifetimes). The exact expression of $E_\Delta$ grows rapidly at increasing $d$ from 0 at $d = 0$ to being already roughly 2/3 of the value quoted above at $d = 2R$ [32].

This clearly gives an upper limit $\tilde{m}$ to mass to leave $\tau$ large enough for the above. This can easily be estimated as follows. If we take the separation as short as $d = 2R$, corresponding to have the two superposed mass distributions on the verge of overlapping, we must have

$$2R = d < \frac{3}{2} \cdot \frac{5}{6} \frac{R}{m^2},$$

which gives $m < \tilde{m} = \sqrt{5/8} = 0.79 \, m_p$, inserting, explicitly, the Planck mass. Any larger $d$ at mass fixed means a larger recombination time and, in addition, a smaller $\tau$; for any given mass, the best option to obtain recombination time $< \tau$ is then to choose $d = 2R$, and the just-given $\tilde{m}$ is the largest allowed mass to have this inequality satisfied.

There is clearly a tension between the collapse models on one side and the possibility to obtain quadrupole emission from recombination on the other. When $m_A$ is, indeed, large enough to allow, in principle, for emission ($m_A > m_{\text{emit}} = m_p$), the collapse models foresee it to decay before it can recombine (and, if we read this the other way around, the delocalization itself of such an $m_A$ is problematic in the first place, with $m_A \approx m_p$ playing, then, the role of an upper-limit mass scale for delocalization to possibly happen in collapse models, cf. [35]). If the proposal of Diósi and Penrose (in its basic form) is correct, there is no possibility to obtain (quadrupole) emission while recombining $A$; this, whichever $m_A$ is and however small we (consistently) take the recombination time $T_A$.

Something that is a little bit striking is the coincidence $m_{\text{emit}} \approx \tilde{m}$ between the (lower-limit) threshold mass $m_{\text{emit}}$ for quadrupole emission from recombination and the (upper-limit) threshold mass $\tilde{m}$ to have the collapse proposal allowing for the delocalized particle to have enough time to recombine (and have it delocalized in the first place). Things happen as if when circumstances would finally allow for emission (delocalized masses large enough), right then the latter is inhibited by the collapse.

As for the paradox, one thing that the consideration of collapse models adds is that if Diósi and Penrose are right, the crucial case $m_A \gg m_p$ can never happen. This immediately means that no paradox can arise (in particular, no need to invoke graviton emission), and, in addition, looking at (15) and (1), that long integration times are needed for $B$ to possibly obtain which-path (this being hardly compatible with a noncollapsing $A$ [4]).

## 4. Summary and Conclusions

We tried to determine the conditions for graviton emission from recombination of a delocalized particle. This was carried out having as background the gedanken experiment [15,16] (in which Alice recombines a delocalized particle ($A$) while Bob tries to perform which-path a distance $D$ apart with a test particle ($B$); in this, a tension between causality and complementarity might potentially arise when Alice and Bob act in times $T_A, T_B < D$ if we assume that the gravitational field sourced by $A$ entangles with the superposed locations).

To this aim, we simply explicitly computed, for generic geometric conditions, the variation of quadrupole moments (of the delocalized particle and its lab, which we called Alice's system) from before to after $A'$ recombination, both in case the field is entangled with the positions and in case it is not and is instead sourced by the expectation value of energy–momentum on the delocalized state. In view of the gedanken experiment, we also computed the difference between the quadrupole moments of the superposed configurations.

We found that the variation of the moments in the recombination is greatly enhanced in case the field is entangled compared to if it is instead sourced by the energy–momentum on the delocalized state (in which case the variation is simply $\sim m_A d^2$, i.e., what is naively expected on dimensional grounds) and provided the gain. We provided a formula for how quickly recombination must occur for graviton emission to set in. In it, the threshold time for graviton emission grows as $m_A$ in place of $\sqrt{m_A}$ (which is what is obtained instead if the variation of quadrupole moment is $\sim m_A d^2$). In all this, graviton emission is found to be possible only when $m_A > m_p$ for recombination times short enough, meaning that for masses smaller than the Planck mass, no graviton emission is possible, however small we (consistently) take the recombination time.

Concerning the gedanken experiment, from the computed difference of the moments in the superposed configurations, we find that a potential clash between causality and complementarity is, in principle, conceivable only when $m_A \gg m_p$ (which comes from requiring Bob to be able to perform which-path in $T_B < D$). Clearly, no clash can arise, however, since for these masses, if $T_A < D$, Alice's system necessarily emits, and the coherence of $A$ is affected without need of causal relationship with $B$, along the lines of [16,18,19]. If the coherence of $A$ is probed, in particular, through inspection of interference fringes when $A$ is recombined, the condition for the onset of emission turns out to coincide with the condition of the separation $\delta$ of the fringes to become $\delta < l_p$, so that the two conditions of the onset of emission on one side and the disappearing of the interference pattern (at ideal conditions) on the other, do result as equivalent in this setting. If, instead, the probing of the coherence of $A$ is performed in another manner (as proposed in [19]) not relying on the detection of the interference pattern, it seems crucial that graviton emission sets in to avoid any clash between causality and complementarity.

This brings with it that if the communication channel is assumed to be nonlocal—instead of local, as implicit in discussion above concerning emission—yet causal, as contemplated (together with the local channel) in [13], it is not so clear how to avoid the paradox when $m_A \gg m_p$ in the noninterferometric setting of [19], since we do not have interferometric fringes to wash off (with finite limit $l_p$), nor we can rely on emission for having $A$ to decohere while recombining it in $T_A < D$; yet performing which-path of $B$ is ideally possible within $T_B < D$, this potentially clashing with complementarity.

When all this is considered within the collapse models of Diósi and Penrose [29–31] (in their basic formulation), we saw that the case $m_A \gg m_p$ can never happen (since the delocalized state decays before it recombines, or before it can be formed in the first place),

and then no paradox can arise since Bob will never be able to perform which-path in $T_B < D$.

Indeed, in these models it is not possible to have $A$ delocalized, even when, simply, $m_A > m_p$. Connected to the above, this means that (quadrupole) emission from recombination would be never possible in them. More precisely, we have the curious coincidence $m_{\text{emit}} \approx \tilde{m} \; (\approx m_p)$ between the threshold mass $m_{\text{emit}}$ for emission and the threshold mass $\tilde{m}$ to have separation that withstands decay, meaning that right when $m_A$ would be large enough to obtain emission (with $A$ recombining in a time as short as possible), it would then become also too large to have $A$ not collapsed yet in one of the two locations.

In closing, we would like to make a comment on the role of Planck length $l_p$ in the above. We saw that the onset of graviton emission in the recombination of a delocalized particle and the washing out of the pattern in an interferometric setting due to the limit $l_p$ are two sides of the same coin. This may lead one to suspect that the existence of a limit length alone, when suitably introduced in the formalism, might account for a great deal of results concerning quantum features of curvature (cf. [36]), this clearly irrespective of the actual underlying quantum theory of gravity.

The systematic investigation of all the consequences of a limit length is the goal of the framework [37,38] (called minimum-length or zero-point-length metric or qmetric), which computes the distance between two points, $p$ and $P$, with a lower-limit-length built in, thus with smallest-scale nonlocality embodied in the biscalar, which provides distances. In this, tensors are replaced by bitensors as fundamental objects in the description, with some selected ones playing a major role. In particular, the metric tensor is replaced by a (qmetric) bitensor which, consistent with the need to provide a finite limit length, diverges in the coincidence limit. We might speculate that the loss of coherence of particle $A$, as described here at weak-gravity conditions (we use Newtonian gravity), might be reobtained as an effect of the qmetric associated with Minkowski (that is, replacing Minkowski with qmetric Minkowski), regardless of graviton emission; a hint, in this sense, might be that, assuming gravity has mediators, the threshold mass for graviton emission we obtained turns out to be the Planck mass (this would cure, by the way, the problem mentioned above of how to avoid a clash between causality and complementary in case we lack interferometric fringes to wash off or gravitons to emit).

On a parallel side, some intriguing curvature-related quantum effects investigated through the use of key bitensors are discussed in [39,40]. In the qmetric, a number of results have been obtained relating curvature, and the dynamics (field equations), to an underlying quantum structure of spacetime (see [41,42]); attempts to investigate the latter are detailed in [43–45].

**Funding:** This work was supported in part by INFN grant FLAG.

**Data Availability Statement:** All data generated or analysed during this study are included in this published article.

**Acknowledgments:** I'm grateful to Alessio Belenchia for useful comments on an earlier version of the paper.

**Conflicts of Interest:** The author declares no conflict of interest.

## Appendix A. Evaluation of Gravitational Gradients and Their Variations

We address the problem of determining the difference of the gravitational gradients felt at a distance in the two configurations corresponding to the two superposed positions 0 and 1 of $A$ (Figure 1). No dipole term can contribute to this difference [16]; the dipole term taken with respect to the center of mass of Alice's system ($A$ + lab of Alice) is actually vanishing in any configuration. In the circumstances assumed in [20] (centers of mass of $A$ and of the lab of Alice coinciding for undelocalized $A$), the quadrupole moments in the two configurations are equal and cannot affect the difference. We claim here that if we consider the slight generalization of not-coinciding centers of mass, the quadrupole

moments with respect to the center of mass of Alice's system are different in the two cases, and they become the dominant contribution, as in [16]. Moreover, their variations result as much bigger in case the field is entangled with the superposed locations than if it is instead sourced by the expectation value of the energy–momentum tensor on the delocalized state.

To see how this comes about, let us write the gravitational potential $\phi$ at a point of spatial coordinates $x^i$, $i = 1, 2, 3$ with respect to some origin, as (cf. e.g. [46])

$$\phi = - \left( \frac{M}{r} + \frac{d_j n^j}{r^2} + \frac{Q_{ij} n^i n^j}{2r^3} + \cdots \right), \tag{A1}$$

where $M$ is the mass of the body which is the source of the potential (in our case, Alice's system: $A$ + lab of Alice), $n^i = x^i / r$ with $r$ the distance to the origin, $d^i$ is dipole moment, and the quadrupole is

$$Q_{ij} = \int \left( 3x'_i x'_j - r'^2 \delta_{ij} \right) \rho \, dV, \tag{A2}$$

where the integral runs over the body, with $r'$ the distance to the point with the attached volume element $dV$ at coordinates $x'^i$, and $\rho$ the density there.

We decide to compute $\phi$, taking as origin the center of mass $C$ of Alice's system. Clearly, this implies that $d^i = 0$, $i = 1, 2, 3$. Now, at points along the $x$ axis, taken as the direction connecting the superposed positions, we have

$$Q_{ij} n^i n^j = \int \left( 3x'^2 - r'^2 \right) \rho \, dV \equiv Q_{xx}. \tag{A3}$$

If we consider the approximation of a mass distribution $\rho_A$ of the $A$ particle given by a Dirac's $\delta$ (for the sake of simplicity, but this can be relaxed), and assume that the particle has coordinate $x_A$ with respect to $C$, we obtain

$$\begin{aligned} \int \left( 3x'^2 - r'^2 \right) \rho_A \, dV &= \int \left( 3x'^2 - x'^2 \right) m_A \, \delta(x' - x_A) \, dx' \\ &= 2\, x_A^2 \, m_A. \end{aligned} \tag{A4}$$

Considering the mass distribution of the lab of Alice (meant specifically as the system of Alice with $A$ removed), and calling $x_{labA}$ the $x$-coordinate of its center of mass with respect to $C$, we have

$$\begin{aligned} \int \left( 3x'^2 - r'^2 \right) \rho_{labA} \, dV &= \int \left( 3x'^2 - x'^2 \right) dM_A \\ &= \int 2 \left( x' - x_{labA} + x_{labA} \right)^2 dM_A \\ &= 2 \left[ \int \left( x' - x_{labA} \right)^2 dM_A + \eta \, x_A^2 \, m_A \right], \end{aligned} \tag{A5}$$

with $M_A$ the mass of the lab, $\eta \equiv m_A / M_A$, and, of course, $x_{labA} = -\eta \, x_A$.

Let us consider general circumstances in which the position of $A$ when not delocalized is not coinciding with $C$ but has, instead, a slight offset $\bar{x}_A \ll r$ along the $x$-axis ($\bar{x}_A = 0$ in the circumstances of [20]). In the configuration of Alice's system corresponding to the particle $A$ in path 0 (see Figure 1), we have $x_A^{(0)} = \bar{x}_A + d/2$, where the index $^{(0)}$ tags the configuration. Analogously, $x_A^{(1)} = \bar{x}_A - d/2$.

Calling $Q_{xx}(0)$ and $Q_{xx}(1)$ the corresponding quadrupoles, from (A3) we obtain

$$Q_{xx}(0) = 2 \left( \bar{x}_A + \frac{d}{2} \right)^2 m_A + 2 \left[ \int \left( x' - x_{labA}^{(0)} \right)^2 dM_A + \eta \left( \bar{x}_A + \frac{d}{2} \right)^2 m_A \right] \tag{A6}$$

and

$$Q_{xx}(1) = 2\left(\bar{x}_A - \frac{d}{2}\right)^2 m_A + 2\left[\int (x' - x_{labA}^{(1)})^2 \, dM_A + \eta \left(\bar{x}_A - \frac{d}{2}\right)^2 m_A\right] \qquad \text{(A7)}$$

with $x_{labA}^{(0)} = -\eta \, x_A^{(0)}$ and $x_{labA}^{(1)} = -\eta \, x_A^{(1)}$. The two integrals here depend only on the *form* of mass distribution of the lab around its actual center of mass in the two configurations; their difference, as well as their difference with respect to the value $Q_{xx}(f)$ for the final configuration with particle $A$ recombined (i.e., $x_A = \bar{x}_A$), can be estimated to be $\mathcal{O}((\eta d)^2 M_A) = \mathcal{O}(\eta \, d^2 \, m_A)$ (and is identically vanishing in the approximation of rigid displacement).

Considering the case of field entangled with the superposed positions, we have that the variations of the quadrupole moments in recombination are $|Q_{xx}(0) - Q_{xx}(f)|$ and $|Q_{xx}(1) - Q_{xx}(f)|$ for positions 0 and 1, respectively, while their difference in the two positions is $Q_{xx}(0) - Q_{xx}(1)$. Neglecting terms containing $\eta$ as a factor, namely, of order $\mathcal{O}(\eta \bar{x}_A^2, \eta \bar{x}_A d, \eta d^2)$, we obtain

$$\begin{aligned} Q_{xx}(0) - Q_{xx}(f) &= 2\left(\bar{x}_A + \frac{d}{2}\right)^2 m_A - 2\,\bar{x}_A^2 m_A \\ &= 2\left(\bar{x}_A d + \frac{d^2}{4}\right) m_A \\ &= Q_A + \frac{d^2}{2}\, m_A, \end{aligned} \qquad \text{(A8)}$$

with $Q_A = 2\,\bar{x}_A d \, m_A$, and analogously,

$$Q_{xx}(1) - Q_{xx}(f) = -Q_A + \frac{d^2}{2}\, m_A. \qquad \text{(A9)}$$

We have, then, $Q_{xx}(0) - Q_{xx}(1) = 2\,Q_A$ and, when $\bar{x}_A$ is significantly larger than $d$ (though still with $\bar{x}_A \ll r$), $|Q_{xx}(0) - Q_{xx}(f)| \simeq |Q_{xx}(1) - Q_{xx}(f)| \simeq Q_A$. This proves Equation (6) and what we said about its meaning in the main text.

We see that, contrary to the case considered in [20], in general, the quadrupoles corresponding to the two configurations are not equal, with a difference $\mathcal{O}(\bar{x}_A d \, m_A)$, which provides the dominant contribution to the difference $\Delta\phi$ in the gravitational field felt by $B$ (of course, in case of ability of the gravitational field to entangle $A$ with $B$).

With this, we can proceed to compute the difference $\Delta x$ in the position of particle $B$ associated with this $\Delta\phi$, assuming $d, \bar{x}_A \ll D$. We have $\Delta\phi = Q_A/r^3 \approx Q_A/D^3$, and then $\Delta g = 3\,Q_A/r^4 \approx 3\,Q_A/D^4$, with $g$ denoting the acceleration of $B$. We then have $\Delta x = 1/2\,\Delta g\,T_B^2 = 3/2\,Q_A/D^4\,T_B^2 \sim Q_A/D^4\,T_B^2$, which, on imposing $\Delta x > 1$, gives Equation (1) in agreement with [16], but with $Q_A$, as in Equation (6).

If the field is not entangled with the positions but is sourced instead by the expectation value of the energy–momentum tensor on the delocalized state, the quadrupole term is not a superposition of two terms but has a well-defined value, and no discrimination is possible between the paths. There is, anyway, a variation $\widetilde{Q}_A$ of the quadrupole moment in the recombination which can be calculated as $\widetilde{Q}_A = \frac{1}{2}[Q_{xx}(0) + Q_{xx}(1)] - Q_{xx}(f) = \frac{1}{2}d^2 m_A$, where $Q_{xx}(0)$ and $Q_{xx}(1)$ are as given in (A8) and (A9). We see this is the same order of magnitude of the naive guess (3) and is, in general, much smaller than $Q_A = 2\,\bar{x}_A d \, m_A$, which is what we obtain instead if the field does entangle with the positions.

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
