# Peer review of "Conditions for Graviton Emission in the Recombination of a Delocalized Mass"

_quantumrep, doi:10.3390/quantum5020028_

Round 1

Reviewer 1 Report

I am not an expert in the field this paper deals. Nonetheless, in my opinion the paper is rather well written and introduces onself in the issue of being sensitive to the quantum nature of gravity with a minimal mathematical formalism.

I have a question regarding the nature of the paradox. How is it possible that the ancillary particle in B would be sensitive to the gravity field entangled with the positions in A but not with that of B if TA and TB are much smaller than D? In standard paradoxical situations one would require first that the quantum system with parts in A and B are entangled and then one would make changes in one of the two parts. I do not see this discussion in the present manuscript.

Other minor points are:

* I find convenient to dedicate some discussion to the meaning of the field states $|\phi_f\rangle$, and why $\langle \phi_1|\phi_0\rangle=\langle 0|\phi_0-\phi_1\rangle$, as written 9 lines below (5).

*  ''We see that (6) gives a same dependence on $m_A$ and
$d$ as (3).'' I find that this sentence is confusing because (6) has a factor $1/d$, such that it is actually linear in $d$, while (3) is quadratic in $d$.

I hope that the author could clarify me these questions before I could recommend this paper for publication in the journal.

Author Response

|I have a question regarding the nature of the paradox. How is it possible that 
|the ancillary particle in B would be sensitive to the gravity field entangled 
|with the positions in A but not with that of B if TA and TB are much smaller 
|than D? In standard paradoxical situations one would require first that the 
|quantum system with parts in A and B are entangled and then one would make 
|changes in one of the two parts. I do not see this discussion in the present 
|manuscript.

Particle A is thought to have been kept in the delocalized state
since long before. If the gravitational field is entangled with
the superposed positions,
when recombination starts and the particle B is released 
the latter feels
a superpositions of gravitational fields
each one corresponding to one of the two branches of the superposition.
(But i'm not quite sure i have grasped the point raised by the referee ..)

I have tried to make this clearer with a change  
in the parenthesis "(the two locations of $A$ ..."
to the left of Fig. 1 at p.2.
(change n.1 in the list of the changes).

|Other minor points are:

|* I find convenient to dedicate some discussion to the meaning of the field 
|states $|\phi_f\rangle$, and why 
|$\langle \phi_1|\phi_0\rangle=\langle 0|\phi_0-\phi_1\rangle$, 
|as written 9 lines below (5).

The context is the linearized quantum description of gravity,
which seems appropriate in our circumstances of weak fields.
The description I do is to use an analogy 
of what happens for the electromagnetic case.

I have tried to explicitly state this in the new version
in the paragraph of Eq. (5),
expanding the description of the fields as well
as the motivation of the equality mentioned by the reviewer.

I have also added three references on this.

These are the changes 2, 3 and 5 in the list.

|*  ''We see that (6) gives a same dependence on $m_A$ and
|$d$ as (3).'' I find that this sentence is confusing because (6) 
|has a factor $1/d$, such that it is actually linear in $d$, 
|while (3) is quadratic in $d$.

I agree with the referee which i thank for this pointing out this.
What I would convey is that in (6) we can still see a dependence
on the quantity m d^2, 
a quantity which is the result of $Q_A$ in (3), 
but with a factor in front which can be much larger than 1. 

I have tried to clarify this in the new version.
This is change n.4 in the list.

|I hope that the author could clarify me these questions 
|before I could recommend this paper for publication in the journal.

Reviewer 2 Report

see attached file

Author Response

I would like to thank the Reviewer for the interest and appreciation of this work. 

Round 2

Reviewer 1 Report

The author have answered my question and I recommend the paper for publication now.